# Knowledge, Attitudes, and Practices of Infection Prevention and Control Nurses in Public Hospitals in the Limpopo Province: A Qualitative Study

**DOI:** 10.3390/ijerph22010116

**Published:** 2025-01-16

**Authors:** Lebitsi Q. Ranoto, Cairo B. Ntimana, Pamela Mamogobo, Eric Maimela

**Affiliations:** 1Department of Public Health, University of Limpopo, Sovenga St., Polokwane 0727, South Africa; komapelq61@gmail.com (L.Q.R.); pamela.mamogobo@ul.ac.za (P.M.); eric.maimela77@gmail.com (E.M.); 2DIMAMO Population Health Research Centre, University of Limpopo, Sovenga St., Polokwane 0727, South Africa

**Keywords:** infection prevention and control, nurses, qualitative study, resource allocation, training

## Abstract

A crucial aspect of delivering healthcare is infection prevention and control (IPC), especially in public hospitals where the high volume of patients and limited resources can heighten the risk of healthcare-associated infections. This qualitative study explores IPC nurses’ knowledge, attitudes, and practices in public hospitals within the Limpopo province of South Africa. The study adopted a qualitative descriptive design. This qualitative study utilized self-developed validated semi-structured interviews with IPC nurses from 12 public hospitals (4 tertiary, 4 regional, and 4 district). The interviews were transcribed verbatim and analysed using thematic analysis to identify key themes related to knowledge, attitudes, and practices in IPC. Each interview lasted approximately 15 to 20 min. Themes and subthemes that emerged provided a structured overview of the key aspects discussed. Each theme captures a different facet of the experiences, perceptions, and challenges faced by IPC nurses in their role. The subthemes further break down these views into specific areas of focus, offering deeper insights into the nurses’ experiences of their professional responsibilities. This study shows that, although IPC nurses have a good understanding of infection control and a positive attitude toward it, systemic problems and resource constraints make it difficult to consistently implement optimal practices. Affective mood, opportunity cost, coherence of the intervention, burden, perceived efficacy, self-efficacy, and ethics are among the major themes that were found. To improve IPC efforts, there is a clear need for more focused training, resources, and managerial support.

## 1. Introduction

A vital part of providing healthcare is infection prevention and control (IPC), particularly in public hospitals where a high patient load and limited resources might increase the risk of healthcare-associated infections (HCAIs) [1,2,3,4]. High-cost healthcare, longer hospital stays, and greater morbidity are all consequences of healthcare-associated infections (HCAIs) [1,5,6,7]. It takes effective IPC programs to reduce these hazards and guarantee patient safety [8].

Infection control procedures are crucial to implement and maintain, and IPC nurses are essential to this process [9,10,11]. They are in charge of managing outbreaks, keeping an eye on compliance with infection control procedures, conducting infection rate surveillance, and training healthcare personnel [12]. Their expertise and meticulous attention to detail are critical to preventing the spread of infections within healthcare facilities, particularly in regions like Limpopo, where healthcare resources are often stretched.

Despite its importance, IPC implementation in Limpopo’s public hospitals faces significant barriers [12,13,14,15]. These challenges include insufficient managerial support, inadequate staffing, limited access to essential supplies, and a lack of comprehensive training programs for healthcare workers [16]. Additionally, the rapid turnover of patients and persistent resource constraints further complicate the effective execution of IPC strategies [3,17]. These issues not only undermine the efficacy of IPC programs but also contribute to the prevalence of HCAIs, posing risks to both patients and healthcare workers [18,19].

This study aims to enhance IPC strategies and improve patient outcomes in Limpopo’s public hospitals by exploring the knowledge, attitudes, and practices of IPC nurses. By examining their perspectives and experiences, this research seeks to identify the barriers they encounter and the factors that support or hinder effective IPC practices. The insights gained will be instrumental in addressing these challenges and tailoring interventions to the unique needs of the region.

Ultimately, this study aspires to strengthen IPC practices, reduce the burden of HCAIs, and contribute to the formulation of more effective policies and programs. These findings will provide actionable recommendations for legislators, hospital managers, and healthcare providers in Limpopo, fostering safer healthcare environments for both patients and healthcare workers while improving overall patient care in the region.

## 2. Materials and Methods

### 2.1. Study Setting

This study was conducted in the Limpopo province, which is named after the Limpopo River which runs at the borders of South Africa with Zimbabwe and Botswana. The Limpopo province is the northernmost province in the country (Limpopo Department of Health Annual Report 2020). Limpopo is divided into five districts, namely the Capricorn District, the Vhembe District, the Waterberg District, and the Mopani and Sekhukhune Districts, and borders with the countries of Botswana to the west, Zimbabwe to the north, and Mozambique to the east. The Beit-Bridge border post (going into Zimbabwe) is the largest border in the province and is considered the gateway to the rest of Africa.

### 2.2. Study Design

This study adopted a cross-sectional, descriptive design. This research design featured a flexible methodology, generating high-quality data which allowed for a more profound insight into lived experiences [20], specifically the knowledge, attitudes, and practices of infection prevention and control nurses working in the IPC program in public hospitals of the Limpopo province. This study was approved by the Turfloop Research Ethics Committee (TREC) with proposal number TREC/111/2023:PG and the Department of Health in the Limpopo province, with reference number LP_2023-04-026.

### 2.3. Population and Sampling

The study population consisted of 12 infection prevention and control nurses from Limpopo public hospitals. The participants were selected through purposive sampling to ensure a diverse representation of experiences and perspectives. Only IPC nurses were considered in the current study.

### 2.4. Data Collection and Measurements

This qualitative study employed self-developed, validated, and semi-structured interviews to explore the knowledge, attitudes, and practices of infection prevention and control (IPC) nurses across 12 public hospitals, including 4 tertiary, 4 regional, and 4 district facilities. Data collection was conducted face-to-face between September and November 2023, using an interview guide designed to elicit in-depth insights.

Three principal authors—LQR, CBN, and EM—who are experienced qualitative researchers, facilitated the interviews. Prior to each interview, the researchers explained this study’s purpose, the voluntary nature of participation, and the right to withdraw at any point without consequences. Written consent was obtained from all the participants. The semi-structured format allowed flexibility while maintaining consistency across interviews. Each session lasted approximately 15 to 20 min and was audio-recorded. The researchers also used notepads to document demographic details. Data collection continued until data saturation was reached, which occurred with participant number 12, ensuring that no new information or themes were emerging by that point.

### 2.5. Data Analysis

The data analysis utilized Tesch’s open thematic approach, applied at multiple levels using text data provided by the patients [20]. Two authors (C.B.N. and E.M.) organized the dataset, which included verbatim transcriptions of all recorded audio files and accompanying field notes. The transcripts were anonymized before entering them into QSR Vivo 10 (QSR International, Warrington, UK) to aid in the analysis. They thoroughly reviewed the transcripts to gain a comprehensive understanding of the material. The next step involved coding the data by identifying meaningful segments and assigning appropriate code labels. These codes were then examined to develop and articulate emerging themes. A collaborative discussion with the other authors was conducted to refine and finalize these themes. The findings, summarized in Table 1, were reported as qualitative results, with all authors reaching a consensus on the themes. Tesch’s open thematic approach offered a systematic framework for coding and categorizing the data, enabling the authors to deeply engage with the dataset and allow new insights to naturally surface. The resulting themes are detailed in Table 1.

### 2.6. Measures to Ensure Trustworthiness

To ensure the accuracy and rigor of the study, the researchers adhered to Lincoln and Guba’s four evaluation criteria, which provide a structured framework for assessing the trustworthiness of qualitative research [20]. Credibility was established through a member-checking process involving feedback from three participants and the authors themselves, ensuring the accuracy of the interpretations. Sufficient time was dedicated to data collection, analysis, and engagement with the data, enhancing the study’s validity. Transferability was addressed by providing detailed descriptions of the research setting, participant demographics, and methods, allowing readers to assess the applicability of the findings to other contexts. The inclusion of participants with diverse characteristics and the use of rich, illustrative quotations further strengthened the contextual relevance of the findings. To ensure confirmability, the researchers maintained an audit trail documenting the study’s procedures, decisions, and adjustments, enabling verification of the findings. Consistency was enhanced by standardizing the interview process, ensuring that all participants were asked the same set of questions. Dependability was achieved through the transparent documentation of all research activities and rigorous review by additional researchers, providing an external check on the accuracy and reliability of the process.

## 3. Results

Table 1 below organizes the themes and subthemes that emerged, providing a structured overview of the key aspects discussed. Each theme captures a different view of the experiences, perceptions, and challenges faced by IPC nurses in their roles. The subthemes further break down these views into specific areas of focus, offering deeper insights into the nurses’ experiences of their professional responsibilities.

Table 2 below provides an overview of the participants involved in evaluating infection prevention and control programs, showing equal representation across regional, district, and tertiary hospitals (33.3% each). Most participants were female (66.7%), with males comprising 33.3%. The age distribution was varied, with the majority aged 51–60 (50%), followed by smaller proportions of younger age groups. All participants were registered nurses, highlighting their critical role in infection control initiatives. Experience in the program spanned from 2005 to 2019, with each year accounting for 16.7%, indicating diverse tenure levels. Educational qualifications were evenly split between certificates, diplomas, and unspecified credentials (33.3% each), reflecting a range of professional training backgrounds.

### 3.1. Affective Attitude

The following statements provide insights from infection prevention and control (IPC) nurses working in healthcare facilities, highlighting their experience. They reference key milestones: the year 2019, nearly five years ago, and the thirteenth year since 2005. Additionally, they highlight the IPC nurses’ tenure, with some having worked at their respective hospitals since 2011—marking 11 years, soon to be 12.


*……“Since 2019 is almost. It’s almost five years. Yeah, four years. in Letaba Regional Hospital.”*
(Regional Hospital IPC Nurse)


*……“This year it will be the. 13 years Louis Trichardt District Hospital.”*
(District Hospital IPC Nurse)

The challenges and benefits of reporting structures for IPC nurses were also emphasized. Reporting directly to the CEO or Nursing Manager offered visibility and underscored the critical role of IPC in enhancing patient safety. However, it also came with difficulties, such as the limited prioritization of IPC concerns. The nurses’ advocated for recommendations aimed at improving patient outcomes and reducing healthcare-acquired infections. Their proposals focused on bridging gaps in knowledge and fostering stronger support for IPC among healthcare professionals and leadership teams.


*……“So with us we report both to the next to deputy nursing manager and in other areas it is hard to manage.”*
(IPC Nurse District Hospital)


*……“No. We report directly to the CEO or the nursing manager and we do have challenges when addressing IPC issues, as they are not prioritized.”*
(IPC Nurse Regional Hospital)

### 3.2. Opportunity Cost

During the interviews, the IPC nurses frequently highlighted and expressed concerns related to the opportunity cost of allocating resources—time, labour, and money—to enforce discipline for non-compliance. 

They highlighted that these resources could be used for other important aspects of IPC, such as training, surveillance, or improving infrastructure.


*……“Shortage of staff. Because if the person that we has been given to screen on daily basis not is not available or she is on leave, there is no replacement. So it’s a challenge and then the other thing is the non-compliance. If you don’t comply with the standard precautions, there is nothing that can be done.”*
(District Hospital IPC Nurse)


*……“I highly recommend the training because from the Infection Control program. It’s not anybody who can do it ready to face the challenges we encounter.”*
(Tertiary Hospital IPC Nurse)

### 3.3. Intervention Coherence

It was felt that maintaining intervention coherence would improve infection prevention and control procedures and results by raising the probability of successful implementation. In the context of infection prevention and control (IPC) programs, intervention coherence is crucial for ensuring that the strategies and actions implemented are logical, understandable, and perceived as effective by those involved. By ensuring intervention coherence, IPC programs can increase the likelihood of successful implementation and ultimately improve infection prevention and control practices and outcomes.


*……“A shortage that we’re experiencing now is really very, very, very difficult.”*
(District Hospital IPC Nurse)


*……“I think maybe the challenge is that that is not recognized as one of the specialties. If maybe that then maybe if it can be recognized as a especially, maybe that’s the way maybe healthcare professional health and also management can take the IPC into consideration.”*
(Regional Hospital IPC)


*……“Shouldn’t allocate one person to the Infection Control program. The program is highly demanding. On a daily basis, you need to go to the ward round, and look at the size of our hospital.”*
(Tertiary Hospital IPC Nurse)

### 3.4. The Burden

According to the hospital IPC nurses’ comments, they frequently have a heavy job that includes conducting monitoring, putting control measures in place, educating patients, and making sure that protocols are followed. During times of high demand or outbreaks, this workload may become worse. Their comments also made it abundantly evident that these nurses have a major duty to safeguard patients and healthcare personnel by preventing healthcare-associated infections (HAIs). This role can be onerous, particularly when there are obstacles to overcome like staff resistance or scarce resources


*……“Because most almost every month we do have the healthcare-acquired infections, most specially the surgical site infections. That’s the most challenging that we are having.”*
(Tertiary Hospital IPC Nurse)


*……“I would say the minimum is 2 and the maximum can be visible 5 or 6 every month.”*
(Regional Hospital IPC Nurse)


*……“I think in the burden is very huge because most of our patients we don’t manage them in our institution and we do not complete their management. Our hospital is very small. So it’s a District Hospital, we usually see the patient and transfer them to other institutions.”*
(District Hospital IPC Nurse)

### 3.5. Perceived Effectiveness

Hospital IPC nurses’ comments made it clear that a program’s perceived effectiveness might provide important new information about areas in need of improvement, enhancing the overall effectiveness of IPC practices.


*……“So it’s a challenge and then the other thing is if you don’t comply with the standard precautions, there is nothing that can be done. IPC nurse will just sing and sing, but there is no measures taken to say you didn’t adhere with the standard precaution. So this time if we are doing, maybe something like a disciplinary measure from higher authority.”*
(District Hospital IPC Nurse)


*……“Mainly the incorrect use of PPE and the non-compliance to hand washing and hand hygiene activities.”*
(Tertiary Hospital IPC Nurse)


*……“We can minimize the infection transmission in the health care settings through the implementation of the all the standard precautions. For example, the implementation of hand hygiene. When healthcare workers are implementing the five moments of hygiene according to their area of the settings of their work and the other thing, the use of PPE, implementing all the rules of the use of the PPE.”*
(Regional Hospital IPC Nurse)

### 3.6. Self-Efficacy

According to the IPC nurses, self-efficacy might affect their choices, behaviours, and general efficacy in preventing and managing infections. Their answers demonstrated how comfortable they are performing IPC procedures such as environmental cleaning, isolation procedures, and hand hygiene


*……“Like now, what we do on daily we are monitoring compliances as far as hand hygiene. And medical concern.”*
(Tertiary Hospital IPC Nurse)


*……“What I’ve seen, is the issue of people not observing health hygiene. You know, even if there are hand hygiene facilities that people tend to, to no, to wash their hands.”*
(Regional Hospital IPC Nurse)


*……”And again, we do have visitors, remember in our institution. Visitors, sometimes they are not, some they are, they are, they are the. They are giving us problems. We once had a visitor, who is not supposed to sanitize according to her culture, as she is a traditional healer. But otherwise, we to try to explain so about the importance of sanitizing and otherwise, in the end she managed to sanitize.”*
(District Hospital IPC Nurse)

### 3.7. Ethicality

The majority of their comments emphasized how IPC nurses uphold moral precepts like beneficence and non-maleficence by placing a high priority on patient safety and well-being. They work to ensure that resources are distributed fairly and equally and that everyone has access to care, including infection control measures


*……“The only problem is that they taken from us the only doctor who had public health background, from us as we speak we don’t have any doctor who we speak to when we encounter challenges.”*
(Tertiary Hospital IPC Nurse)


*……“The only challenge that we have observed is that doctors are no longer like the doctors that we knew.”*
(Tertiary Hospital IPC)


*……“Whereby the doctor will prescribe an antibiotic now, and tomorrow when doing ward rounds, he checks the prescription and see if patient received all doses that he prescribed.”*
(Tertiary Hospital IPC)

## 4. Discussion

This study reveals that, although IPC nurses have a good understanding of infection control and maintain a positive attitude toward it, systemic problems and resource constraints hinder the consistent implementation of optimal practices. Despite their knowledge and dedication, IPC nurses face significant challenges that impede their ability to uphold high infection control standards. A lack of essential resources, such as personal protective equipment (PPE), disinfectants, and hand sanitizers, emerged as a major challenge in this study. This finding is consistent with Wee et al. [21], who documented similar shortages during the COVID-19 pandemic, leading to compromised infection control measures in healthcare facilities. The scarcity of supplies often forces IPC nurses to prioritize their efforts, leaving some areas inadequately managed. This prioritization, while pragmatic, is noted by Allegranzi et al. [22,23] as a common issue in low- and middle-income countries, where financial constraints further exacerbate the challenge of resource allocation.

Inadequate staffing was identified as a critical issue, contributing to an overwhelming workload for IPC nurses and limiting their ability to perform key functions such as compliance monitoring, education, and surveillance. This finding aligns with a study by Mitchell et al. [24], who demonstrated a direct correlation between nurse staffing levels and hospital infection rates. Similar patterns have been observed in other resource-limited settings, where high patient-to-nurse ratios and staff shortages negatively impact the effectiveness of IPC measures [10,24].

While the IPC nurses in this study were knowledgeable, limited opportunities for continuous education and professional development were noted as barriers to staying current with evolving best practices. In agreement with the findings of the present study, previous works highlighted similar challenges, emphasizing that resource constraints often restrict access to ongoing training programs [25,26,27]. In contrast, settings with structured training opportunities have demonstrated improved IPC outcomes, as continuous education equips healthcare workers with the skills needed to address emerging infectious threats [28].

The findings also revealed a lack of managerial support, which hindered the implementation and enforcement of IPC measures. This aligns with Magadze et al. [10], who identified management support as a critical factor in the success of IPC programs. Without managerial engagement, IPC nurses struggle to secure the necessary resources and foster a culture of accountability. The existing literature consistently emphasizes the importance of leadership in driving compliance and prioritizing infection control within healthcare facilities.

Financial limitations were a recurring theme in this study, restricting the procurement of essential supplies, investment in advanced technologies, and the expansion of training programs. This is consistent with Allegranzi et al. [22] and Rosenthal et al. [29], who indicated that underfunded healthcare systems often struggle to implement effective IPC measures. Financial constraints not only hinder operational efficiency but also weaken the overall sustainability of IPC initiatives.

Outdated hospital infrastructure, such as poor ventilation, overcrowded wards, and inadequate sanitation facilities, emerged as a significant barrier to effective IPC practices. These findings are in agreement with other studies that reported that infrastructure deficiencies are a common challenge in low- and middle-income countries [30,31,32,33]. The modernization of infrastructure is essential for creating environments conducive to effective infection control, yet it remains a persistent issue in resource-limited regions like rural South Africa.

Time constraints were another challenge identified in this study, with IPC nurses struggling to balance competing priorities such as surveillance, education, and follow-up investigations. Mitchell et al. [24] similarly found that time limitations significantly impact the thoroughness of IPC activities. In Limpopo, this issue is compounded by a high patient turnover and multitasking demands, reducing the time available for focused IPC interventions. In summary, this study corroborates findings from the existing literature, highlighting how systemic barriers, such as resource shortages, staffing constraints, inadequate training, and infrastructure challenges, impede the effectiveness of IPC practices in resource-limited settings. By directly comparing these findings with prior research, it becomes evident that addressing these barriers requires comprehensive, multi-level interventions tailored to the specific needs of regions like Limpopo.

### Study Limitation

This study’s findings may be limited by a moderately small sample size, which could restrict the generalizability of the results to all IPC nurses in Limpopo or similar settings, as well as by potential gaps in capturing the full range of experiences across different hospital types.

## 5. Conclusions

The present study highlights the critical role of IPC nurses in maintaining public health within hospital settings. Despite their extensive knowledge and positive attitudes toward infection control, systemic challenges and resource limitations significantly hinder the consistent implementation of best practices. Enhancing IPC efforts in public hospitals necessitates addressing these issues through targeted training, increased budget allocations, and stronger managerial support. The insights gained from this research provide a foundation for developing more effective IPC strategies tailored to the specific needs of resource-constrained healthcare environments.

Based on the findings of the present study several key strategies should be implemented. First, developing targeted training programs that are regularly updated to reflect the latest best practices in infection control is essential. These programs should be accessible and cover a wide range of topics, from basic hygiene practices to advanced IPC techniques. Second, investing in infrastructure improvements, such as modernizing hospital facilities, upgrading ventilation systems, and improving sanitation facilities, will further support effective infection control and reduce overcrowding in in-patient areas. By implementing these strategies, public hospitals can enhance their IPC programs, ultimately improving patient outcomes and reducing healthcare-associated infections.

Lastly, future research should address the limitations and gaps identified in this study, including evaluating the impact of training programs on IPC efforts in resource-limited hospitals. Such studies can provide evidence-based insights into the effectiveness of educational interventions and their role in mitigating healthcare-associated infections. Additionally, exploring innovative solutions to overcome systemic challenges, such as leveraging technology or community-based support mechanisms, will contribute to the development of more resilient IPC systems in resource-constrained settings.

## Figures and Tables

**Table 1 ijerph-22-00116-t001:** Themes and subthemes identified in the analysis of infection prevention and control (IPC) nurses’ experiences.

Theme	Subthemes
Affective Attitude	Job satisfaction and motivation
	Emotional responses to IPC challenges
	Pride in contributing to patient safety
	Stress or frustration due to non-compliance
Opportunity Cost	Resource allocation challenges
	Time spent enforcing compliance versus other IPC activities
	Balancing enforcement with staff morale and collaboration
Intervention Coherence	Clarity and alignment of IPC guidelines
	Rationale and evidence supporting IPC interventions
	Feasibility and user-friendliness of IPC measures
	Communication and stakeholder engagement
Burden	Increased workload due to IPC responsibilities
	Resource constraints (e.g., staff, funding, materials)
	Emotional and physical fatigue associated with IPC roles
Perceived Effectiveness	Impact on patient outcomes
	Reduction in healthcare-associated infections (HAIs)
	Adherence to IPC policies and practices
Self-Efficacy	Confidence in implementing IPC measures
	Training and knowledge of IPC
	Ability to overcome barriers in IPC practices
Ethicality	Fairness in enforcing IPC policies
	Moral responsibility to prevent HAIs
	Ethical dilemmas in resource-limited settings

**Table 2 ijerph-22-00116-t002:** Distribution of the sample interviewed by selected variables.

Type of Hospital	No	%
Regional	4	33.3
District	4	33.3
Tertiary	4	33.3
Gender		
Male	4	33.3
Female	8	66.7
Age		
30–40	2	16.7
41–50	2	16.7
51–60	6	50.0
61 and above	2	16.7
Position category		
Registered nurse	12	100.0
Years in the program		
Since 2005	2	16.7
Since 2011	2	16.7
2011	2	16.7
2013	2	16.7
Since 2016	2	16.7
Since 2019	2	16.7

## Data Availability

The data presented in this study are available upon request from the corresponding author. The data are not publicly available due to privacy or ethical restrictions.

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
