# Peer review of "Knowledge, Attitudes, and Practices of Infection Prevention and Control Nurses in Public Hospitals in the Limpopo Province: A Qualitative Study"

_ijerph, 2025, doi:10.3390/ijerph22010116_

Round 1

Reviewer 1 Report

Comments and Suggestions for Authors

Thanks for inviting me to review the article titled “Knowledge, Attitudes, and Practices of Infection Prevention and Control Nurses, in Limpopo Province, Public Hospitals, 2024: A Qualitative Study. The article is much needed as the authors evaluate a critical area in healthcare, particularly in resource-limited settings. The objectives are clear. However, several concerns were identified and need revisions.

1.       The introduction section is fairly well-written. But, authors need to write in a logical order and more relevant way, especially to the context of the region (Limpopo).

2.       Methodology is the area I found numerous concerns.

a.       Sample size – Is it sufficient to conclude based on the sample included in the study? I suggest that the authors need to verify the standard sources and cite it accordingly.

b.       Did the authors validate the interview guide?

c.       More details are required about ethical concerns

d.       The authors mentioned the identification of themes and subthemes, but it does not explain how themes were derived or how inter-coder reliability was ensured during the coding process. This weakens the trustworthiness of the findings.

e.       The thematic analysis

3.       The discussion section inadequately integrates findings with existing literature. For example, there is a superficial discussion about identified challenges. However, it is critical to compare and contrast, especially in resource-limited settings.

4.       Again, similar to the introduction, the policy implications also are generic. The authors need to make it clear that conclusions are more relevant to the local settings.

5.   Please include more relvant references to the context of the study.

Wish you all the best  

Comments on the Quality of English Language

Generally well-written. But some areas, especially in discussion section, need improvement. Also, logical sequence is missing.

Author Response

Dear reviewer thank you for your comments  please find the attached 

Reviewer 2 Report

Comments and Suggestions for Authors

Dear authors,

I congratulate you on the manuscript, which is highly relevant to the field of study, as it critically addresses the role of nurses in infection prevention and control, particularly in hospitals with limited resources.

The manuscript is well structured, but requires some important revisions to address identified weaknesses and expand the robustness of the findings, as suggested below:

1. Summary Section:  

Current edition: The abstract does not specify the sample size, which may make it difficult for readers to understand the scope of the study.

Suggestion: Include the sample size (number of nurses) and field of study (number of public hospitals) in the abstract.

Current edition: The methods used for data collection and analysis are not clear from the abstract.  

Suggestion: Provide more details on data collection and analysis methods to improve readers' understanding of methodological rigor.

Current edition: The results are presented in general terms, without highlighting the themes that emerged.

Suggestion: Present the thematic units in the summary.

2.    Introduction Section:

Current edition: The introduction does not present the gaps in the literature that the study under evaluation intends to address.

Suggestion: Clearly present the gaps in the literature and the limitations of studies on the subject in hospitals with limited resources.

Current edition: Superficial presentation of the challenges faced by infection prevention and control nurses. Current epidemiological data to help justify the study are lacking.

Suggestion: Use important and current theoretical references for the topic, which provide more details about the challenges faced by these professionals, thus improving the readers' understanding of the study context. Present current epidemiological data from other studies.

Current edition: The hypotheses and research questions are not clear in the introduction.   

Suggestion: Include the hypotheses and research questions in the introduction.

3.    Materials and Methods Section:

Current edition: Lack of description of possible biases generated by self-reported data and mitigation measures. 

Suggestion: Describe in detail potential biases and how they were mitigated in the data collection and analysis process.

Current edition: The study location is missing. The authors mention the province and districts, but do not provide information on the institutions of the study participants.

Suggestion: Present the number and characteristics of the study sites. General or specialized hospital? How many beds? How many nurses are on staff at these hospitals? Some information is found in the data collection, I suggest reallocating it to the “Study Site” Section.

Current edition: It does not describe the instrument used to collect data, nor how participants will be identified.

Suggestion: Describe in detail the data collection instrument (interview script) and the way in which participants were identified.

Current edition: There is no information on whether there was saturation of the collected data and the definition of the sample size.

Suggestion: Detail the reasons for the number of study participants.

Current edition: The section does not present the ethical considerations of the research.

Suggestion: Present the ethical considerations of research on a specific topic, ensuring transparency.

4.    Results Section:

Current edition: Presence of direct quotes from research participants without identification.

Suggestion: Identify missing citations (page 5, lines 168-169; page 6, lines 199-200).

5.    Discussion Section:

Current edition: Discussions between research findings and existing findings in the literature are weak. 

Suggestion: Include more detailed comparisons, with additional scientific evidence to support interpretations.

6.    Limitations of the study:

Current edition: The authors consider the reliance on self-reported interview data to be a limitation because it introduces the possibility of bias, i.e., participants may provide answers that do not accurately reflect their true perceptions or experiences.

Suggestion: Rewrite this part or remove it, considering that this research method seeks to understand the meaning or internal logic that subjects attribute to their actions, representations, feelings, opinions and beliefs. Conducting the interview relies on the theoretical-methodological knowledge, acceptance and empathy of the researcher, as well as the availability and trust of those who contribute with their testimony.

7.    Conclusion Section:

Current edition: Lack of suggestions for future research.

Suggestion: Add suggestions for future research that address the limitations and gaps identified in this study, such as evaluating the impact of training programs on infection prevention and control efforts in resource-limited hospitals.

References: Only about 65% of the references are from recent studies (last 5 years), so there is a need to replace outdated studies with more recent studies, or to add other current studies. I suggest adjusting the references according to the standards used by IJERPH.

I wish you success with the manuscript.

Author Response

Dear reviewer thanks for the comments please find the attached below 

Reviewer 3 Report

Comments and Suggestions for Authors

Title: The term "2024" in the title, may be unnecessary since the year of the study is typically mentioned in the introduction or methodology.

Abstract: There are redundancies in the abstract, such as "the study adopted a qualitative descriptive design" and "data were analyzed using thematic analysis," which could be consolidated.

Introduction: The introduction could include more examples of effective IPC strategies in low- and middle-income contexts to reinforce the study's relevance.

Methodology: The methodology should justify the sample selection and explain how the validity of the interviews was ensured. It should clarify how inclusion and exclusion criteria for participants were applied. Details regarding data collection and pilot testing procedures should also be provided to validate the interview questions.

Results: The presentation of results could be clearer, particularly in the table of themes and sub-themes (Table 2). While the current categorization is useful, it could be complemented with representative quotes for each theme. Consolidating tables with demographic and outcome data might enhance readability. Additionally, the discussion on differences among nurses at various hospital levels (regional, district, tertiary) could be elaborated.

Discussion: The discussion adequately addresses resource limitations but could delve deeper into the relationship between workload and perceived effectiveness of IPC practices. Practical implications of the results could also be expanded upon, such as:

  • The importance of prioritizing specific IPC training programs;
  • Investments in technology for infection monitoring and control.

Conclusion: The conclusion effectively summarizes the findings but could be more assertive in its recommendations.

Additional Considerations:

  • Including more details on data validation and strategies to minimize biases would enhance the credibility of the results.
  • The article does not provide a detailed explanation of how the sample size was determined. Although qualitative studies do not require formal statistical calculations for sample size, a clearer description of how data saturation was reached would strengthen the methodological rigor.
  • The sample of 12 nurses may not capture the diversity of experiences across different hospital levels (primary, secondary, tertiary). Stratifying by hospital type or role could improve representativeness.
  • The study does not mention measures to mitigate response biases, such as interviewer influence or social desirability bias. These factors could distort the reported perceptions.

Author Response

Dear reviewer thank you for the comments please find the table of corrections below 

Reviewer 4 Report

Comments and Suggestions for Authors

Which software was used to perform qualitative data analysis? MAXQDA ??

Which criterion was used to decide on the sample size? The number of interviewees is quite low. There should be a maximum of 32 people in focus group interviews.

If maximum diversity sampling is taken as a basis; only age, gender and certificate information is available in the nursing group from personal information, there are no other areas of expertise.

Table 1 shows NA values ​​for age and education. If a focus group interview was conducted, NA values ​​should be completed. There should be no unknown values.

For data verification, by using individual interview and observation methods in qualitative research, data triangulation can be performed to increase the accuracy of the studies and the phenomenon studied can be fully understood. Were data triangulation methods used in the study?

Author Response

Dear reviewer thanks for the comments please find the attached table of corrections  

Reviewer 5 Report

Comments and Suggestions for Authors

Dear Authors,

the article is well done, but I think it needs to be improved methodologically. Here my suggestion:

- The introduction is poor, please improve it.

- Please define better the criteria by which the samples were selected. Also, I would ask to define whether the sample is significant or not, otherwise the study may lose its value.

- I would ask to include all questions that have been asked as supplementary material and specify whether they are the same for everyone.

- Overall, the methodology needs to be improved in order to make it more scientific.

Kind regards

Author Response

Daer reviewer thanks for the comments please find the attached 

Round 2

Reviewer 1 Report

Comments and Suggestions for Authors

Good work done by the authors

Author Response

Dear Reviewer.

Thanks you. 

Reviewer 2 Report

Comments and Suggestions for Authors

Dear authors,

The report submitted presents superficial responses to previous suggestions, which made it difficult to identify the corrections made to the manuscript. I suggest that the report suggestion provided by IJERPH be used in a future submission to the journal.

The manuscript still requires minor revisions, as described below:

    1.    Results Section:

Suggestion – Re-edit table 1, on page 4-8, as it is completely misconfigured in the received file.

    2.    Discussion Section:

Suggestion – Add the following sentence at the end of the discussion, in limitations of the study, which were completely excluded in this version of the manuscript: “The study's findings may be limited by a moderately small sample size, which could restrict the generalizability of results to all IPC nurses in Limpopo or similar settings, as well as by potential gaps in capturing the full range of experiences across different hospital types”.

I wish you success with the manuscript.

Author Response

Dear reviewer, thanks for the comments, please find the table of corrections  

Reviewer 5 Report

Comments and Suggestions for Authors

Dear Authors,

Unfortunately, I have doubts about the significance of the sample. You can't believe it is significant, you where to test it.

Kind regards

Author Response

Dear reviewer, Thank you for the comments.
please find the attached table of corrections 
